# Breastfeeding in PKU and Other Amino Acid Metabolism Disorders—A Single Centre Experience

**DOI:** 10.3390/nu16152544

**Published:** 2024-08-03

**Authors:** Agnieszka Kowalik, Sylwia Gudej-Rosa, Marta Nogalska, Joanna Myszkowska-Ryciak, Jolanta Sykut-Cegielska

**Affiliations:** 1Institute of Mother and Child, Department of Inborn Errors of Metabolism and Paediatrics, Kasprzaka 17A, 01-211 Warsaw, Poland; marta.nogalska@imid.med.pl (M.N.); sylwia.gudej@imid.med.pl (S.G.-R.); 2Department of Dietetics, Institute of Human Nutrition Sciences, Warsaw University of Life Sciences (WULS), 02-776 Warsaw, Poland; joanna_myszkowska-ryciak@sggw.edu.pl

**Keywords:** breastfeeding, inborn errors of amino acid metabolism, phenylketonuria, infants

## Abstract

In addition to the numerous immunological and nutritional benefits that breast milk offers to infants, its proportion in the diet must be limited or even excluded in the case of inborn errors of amino acid metabolism (IEM). The objective of the study was to expand knowledge about breastfeeding and the degree of contribution of breast milk to the feeding of infants with IEM before and after the introduction of expanded newborn screening. A retrospective single-centre study was conducted on 127 infants born between 1997 and 2020: 66 with phenylketonuria (PKU), 45 with other IEM (non-PKU), all diagnosed through newborn screening (NBS), and 16 non-PKU diagnosed through selective screening (SS). The time of initiation of dietary treatment and the proportion of breast milk in the diet, both expressed and breastfed, with or without intake control, were analysed at 1, 3, and 6 months after birth. For 47% of the newborns in Groups 1 and 2, the dietary treatment was started before the 10th day of life; in Group 3, the dietary treatment was started after the 10th day of life for all children. During the first month of life, the proportion of infants receiving breast milk was higher in the NBS-PKU (74%) and the NBS non-PKU (80%) groups, compared with 38% in the SS non-PKU infants. In the subsequent months of life, the proportion of infants receiving human milk (either from the breast or a bottle) declined in all groups. This decline occurred more in bottle-fed rather than directly breast-fed infants. Our observations indicate that the model of feeding from a bottle with expressed milk may have had an adverse effect on maintaining lactation and may have contributed to a faster transition to formula milk. Maintaining lactation and extending the period of feeding the infant with human milk in the first 6 months of life is possible by breastfeeding on demand, under regular biochemical monitoring: preferably weekly in PKU infants, and at least every 2–4 weeks in infants with other IEM.

## 1. Introduction

The practice of breastfeeding is a natural and essential one that encompasses both the physical nourishment of the infant and the emotional bonding between the mother and her child. The World Health Organisation (WHO) and other scientific bodies worldwide recommend that mothers exclusively breastfeed their infants for the first 6 months of life to achieve optimal growth, development, and health [1,2]. The European Society for Paediatric Gastroenterology, Hepatology, and Nutrition (ESPGHAN) recommends that infants be exclusively breastfed for a minimum of 17 weeks [3]. 

Human milk is regarded as the “gold standard” for infant nutrition. It is recommended for exclusive breastfeeding for the first 6 months of a child’s life, and then, with its appropriate share in the diet, for a period of two years or longer—as long as this is mutually desired by the mother and the child [1]. It is widely acknowledged that the qualitative composition of human milk is irreplaceable, and that the health benefits of breastfeeding also include long-term effects on cognitive development, behaviour, and mental health in children [4]. Nevertheless, the actual duration of exclusive breastfeeding (EBF) or any breastfeeding (BF) is frequently considerably less than the recommended standards, and still not all infants are breastfed [5]. An even more complicated situation arises when a child suffers from an inborn error of metabolism, and thus, requires a special diet. This can increase the mother’s stress levels, potentially leading to a shorter duration of breastfeeding. The provision of breastfeeding support has been demonstrated to have a significant impact on the total duration of breastfeeding and the risk of the early introduction of infant formula [6]. It is therefore important that mothers and families are fully informed about the benefits of breastfeeding and extended breastfeeding after the introduction of complementary foods. Breastfed infants benefit from the additional protection provided by breast milk, which complements the infant’s innate immunity [7]. The nutrients in the breast human milk adapt to the needs of the growing child, and their content does not decrease with the duration of lactation [8]. The advantages of breastfeeding extend beyond the infant to the mother. These include a lower long-term postpartum weight retention [9], a reduced risk of developing breast cancer in later life [10], and a lower incidence of depression or mood disorders [4].

Breast milk offers a multitude of immunological and nutritional benefits that cannot be replicated by any standard infant formula. Additionally, breast milk contains a plethora of non-nutritional bioactive compounds that may reduce gut propionate production [9,11,12,13], protect against infections [14,15], shape the gut microbiota [16], and improve cognitive development [17]. All these make breast milk an especially beneficial option for infants with inborn errors of metabolism (IEM).

Nevertheless, the proportion of human milk in the diet of children with IEM must be limited in cases of amino acid metabolism disorders, and its complete exclusion from the diet is required in diseases such as galactosemia, abetalipoproteinemia or long-chain 3-hydroxyacyl-coenzyme A dehydrogenase deficiency (LCHAD) and the severe form of very long-chain acyl-CoA dehydrogenase deficiency (VLCAD) [18]. The first publication on breastfeeding in phenylketonuria (PKU) was published in 1981 [19,20], and subsequent decades have seen the accumulation of evidence confirming the safety and health benefits of breastfeeding on demand [21,22,23,24]. 

The implementation of an expanded newborn screening programme has been shown to have a positive impact on the pre-symptomatic detection of other inborn errors of metabolism (non-PKU) and the earlier initiation of appropriate dietary treatment, including the usage of human milk. However, there is a paucity of clinical experience in this group, with only a few studies reporting on the subject. Nevertheless, the authors of these studies recommend breastfeeding despite the difficulties that may be encountered [25,26,27,28]. It is important to highlight that the detailed knowledge of the quantitative composition of human breast milk, including the content of amino acids [29,30], has enabled the control of their supply and dietary intake in patients with inborn errors of metabolism. The growing availability of rapid biochemical tests for assessing metabolic control facilitates the monitoring of dietary management, which in turn enables an increasing number of newborns with inborn errors of metabolism to be fed with human milk. The safe daily intake of breast milk depends on the type of IEM, its clinical severity, the individual’s natural protein tolerance, and the results of current biochemical tests. It is important to note that the goal of dietary treatment for IEM is to maintain metabolic balance and the normal/optimal psychosomatic development of the child. 

This study, based on extensive experience in managing children with inborn errors of metabolism in a reference centre in Poland, aims to expand knowledge about breastfeeding and the degree of contribution of breast milk to the feeding of infants with IEM, particularly the ones identified by expanded newborn screening.

## 2. Materials and Methods

### 2.1. Project Design

A retrospective single-centre study was conducted on 127 infants with amino acid IEM. Patients were divided into three groups: Group 1 (NBS PKU; n = 66)—patients diagnosed through newborn screening (NBS) with classic PKU, born between 2018 and 2020;Group 2 (NBS non-PKU; n = 45)—patients diagnosed through newborn screening with: GA1-12, MSUD-7, PA-1, MMA-5, IVA-12, HCU-3, OTC-2, CIT t.1-2, and ASA-1, born between 2014 and 2020;Group 3 (SS non-PKU; n = 16)—patients diagnosed through selective screening (SS) with: GA1-5, MSUD-3, PA-1, MMA-2, and TYR t.1-5, born between 1997 and 2012.

Patients diagnosed with PKU in NBS remain under the care of the Department of Inborn Errors of Metabolism & Paediatrics at the Institute of Mother and Child in Warsaw, Poland, until they are 6 months old. Following this period, patients are transferred to the Metabolic Outpatient Clinic of the aforementioned institute. The remaining patients from Groups 2 (NBS non-PKU) and 3 (SS non-PKU) remain under the constant care of the Institute of Mother and Child.

Based on medical history, including dietary records, the following were assessed:the number of infants for whom dietary treatment was initiated at <10 and ≥10 days of life in the examined groups;the number of children who were fed with human milk (HM) at 1, 3, and 6 months after birth, divided into breastfed (with or without weighing the infant before and after feeding to control consumption) or bottle-fed with expressed human milk in accordance with the recommended daily intake;the number of children who were not fed with human milk at 1, 3, and 6 months of age.

In the group of infants fed with human milk, five types of diet were distinguished:HM + SF—human milk + special formula (free of amino acid/acids for IEM)HM + IF + SF—human milk + infant formula + special formulaHM + Pf-F—human milk + protein-free formulaHM—human milkHM + IF—human milk + infant formula.

### 2.2. Biochemical Control

Patients with PKU were monitored on a weekly basis, with phenylalanine (Phe) levels being measured using dry blood spots (DBS) by tandem mass spectrometry. In the case of other IEM patients, the biochemical monitoring was adjusted on an individual basis, with plasma measurements typically occurring every 2–4 weeks. Plasma amino acid levels were the basic parameters for metabolic balance evaluation and current diet modification.

The target amino acid levels for PKU and other amino acid disorders were as follows:PKU—Phe—120–360 µmol/L (<12 yrs of age)GA1—normal reference values (optimally in the middle of the range)MSUD–leucine—75–300 µmol/L, isoleucine—200–400 µmol/L, valine—200–400 µmol/L (<5 yrs of age)PA; MMA; IVA—branched chain amino acids within normal reference rangetHCU—<50 µmol/L (children)OTC; CIT t.1; ASA—branched chain amino acids within normal reference rangeTYR t.1–tyrosine—200–400 µmol/L and Phe—>50 µmol/L (<12 yrs of age)

Plasma amino acids were analysed using pre-column derivatisation with o-phthalaldehyde (OPA)/3-mercaptopropionic acid (3-MPA) for primary amino acids, fluorenylmethyloxycarbonyl chloride (FMOC) for secondary amino acids and iodoacetic acid to block the thiol group of homocysteine. The analytes were separated using a C18 column (gemini-NX, Phenomenex) on a Shimadzu Nexera X2 HPLC with gradient elution: eluent A, 40 mm phosphate buffer pH 7.8; eluent B, water/methanol/acetonitrile (10:45:45). Derivatisation reagents were purchased from Sigma-Aldrich. Organic eluents were purchased from Merck.

In “dry” blood spot analysis (collected on PKU-NBS EBF 903 filter paper), amino acids were analysed by LC/MS/MS flow injection as butanol derivatives (MRM scanning in positive polarisation mode). For each of the analytes, specific MRM transitions were used. For quantification, isotopic internal standards of amino acids diluted with methanol were used (NSKAB mixture from Cambridge Isotope Laboratories, Cambridge, MA, USA).

### 2.3. Dietary Assessment

A dietary assessment based on three-day food records was conducted on a weekly basis in patients with PKU, with any necessary adjustments to the diet made in accordance with the obtained phenylalanine (Phe) levels. In the remaining patients (from the NBS non-PKU and SS non-PKU groups), the diet was modified after obtaining the results for amino acid levels in plasma during a follow-up visit (every 2–4 weeks on average) and between follow-up visits (e-mail contact), based on the current body weight, while maintaining the assumed demand and individual tolerance of protein or amino acid intake.

In planning the supply and monitoring the consumption of human milk, national data on the content of amino acids in 100 mL of human milk were used, including: phenylalanine (47 mg), leucine (131 mg), isoleucine (71 mg), valine (72 mg), lysine (86 mg), and methionine (16 mg) [30].

## 3. Results

### 3.1. Characteristics of All Groups 

In 47% of newborns in Groups 1 and 2, dietary treatment commenced prior to the 10th day of life, with the remaining 53% receiving dietary treatment at a later stage. In Group 3, dietary treatment was initiated in all children after the 10th day of their life (Table 1).

Table 2 presents a comparison of the proportion of human milk in the infant diet and the method of feeding (breast or bottle) for each study group. In the first month of life, among the infants who were fed with human milk (n = 91), 47 (37%) received human milk directly from the breast and 44 (35%) from the bottle. In the third month, 42 (33%) infants were still breastfed, while the percentage of infants receiving human milk from a bottle dropped to 12%. At 6 months of age, of the 34 (27%) infants receiving human milk, 24 (19%) were still breastfed, while only 10 (8%) were bottle-fed infants.

### 3.2. Individual Group Characteristics

At 1 month, the proportion of infants receiving human milk was notably high in both the NBS-PKU (74%) and NBS non-PKU (80%) groups, in comparison to the 38% observed in the SS non-PKU group. Latching was possible in 35% of infants in group 1 (NBS PKU) and 49% in Group 2 (NBS non-PKU). Infants in Group 3 were more likely to receive human milk from a bottle (25%) than directly from the breast (13%). In Groups 1 and 2, 39% and 31% of infants received human milk from a bottle, respectively (Figure 1).

In the third month of life, the percentage of breastfed infants remained at a similar level in Groups 1 and 2 as in the first month of life (35% and 38%, respectively), while the number of infants receiving human milk from a bottle decreased significantly (8% and 16%, respectively). The percentage of infants not fed with human milk increased to 58% and 47% in Groups 1 and 2, respectively. In the subsequent months of life, the proportion of infants receiving human milk (either from the breast or a bottle) declined in all groups, with 29% and 31% of infants at 6 months of age in Groups 1 and 2, respectively, receiving human milk. In Group 3, only one patient received human milk.

The analysis of the type of infant diet (Table 3) showed that in Group 1 (NBS-PKU), the HM + SF-based diet predominated throughout the entire observation period. This was observed in 69, 49, and 33% of the infants at 1, 3, and 6 months of age, respectively. The second most common diet in this group was HM + IF + SF. The proportion of IF varied and compensated for the deficit in HM.

In Group 2 (NBS non-PKU), all five types of diet were employed, but until the age of 6 months, the diet consisting of HM + SF was the most frequently used. In the first month of life the proportion of infants receiving human milk (HM) was 42%, while in the third month of life, it was 25%. However, in the sixth month of life it was only three (8%) infants. Human milk was also combined with IF + SF in 17% of infants at 1 month of age and in 8% at 3 and 6 months of age. Human milk combined with protein-free formula was present in the diet of 19% and 17% of infants at 1 and 3 months of age, respectively. Feeding with human milk alone was performed for three infants from Group 2 (two with IVA and one with MSUD) and was continued throughout the first 6 months of life. The HM + IF diet was used for five infants at 1 month of age, but only three and then two infants maintained it at 3 and 6 months of age. The number of all types of diets containing human milk (except HM only) in Group 2 decreased in the 6th month of life by 80% in the case of HM + SF and by 50, 57, and 60% in the case of the HM + IF + SF, HM+ diet Pf-F, and HM + IF, respectively. 

In Group 3 (SS non-PKU), only in the first month of life was it possible to introduce a diet containing HM in two infants combined with SF and in two infants with IF + SF. Two infants were exclusively breastfed in the first month of life, but at 3 and 6 months, only one infant was breastfed.

Breastfeeding without intake control was rare: In Group 1 (NBS PKU) only eight infants were treated this way; in Group 2 (NBS non-PKU) five infants with IVA, one with MMA, and one with MSUD were treated this way. However, in Group 3, two infants were breastfed on demand, but before the diagnosis of the disease.

## 4. Discussion

Due to its nutritional value, breast milk is the optimal food for newborns and infants. The composition of human milk undergoes changes over time, ensuring the proper somatic development of the child by improving brain myelinisation and the peripheral nervous system, balancing the colonisation of intestinal microbiota, and improving physiological immunity. Furthermore, suckling milk from the breast prepares the infant for the consumption of solid foods from a spoon, and facilitates the development of the speech organ in the following months of life. The experiences of breastfeeding in patients with inborn errors of metabolism have been extensively documented in the literature. These include studies on phenylketonuria [21,24,26]. In most cases, infants with PKU who have been diagnosed early are able to continue feeding with human milk. This is facilitated by regular biochemical checks (tandem mass spectrometry), which enable quick modification of the diet, including the consumption of human milk. In our study, the dietary restriction of phenylalanine intake was initiated in accordance with the European Guidelines for PKU [31] before the 10th day of life in 31 (47%) newborns with PKU (Group 1). In the remaining 35 infants from this group, the most common cause of delay in introducing the diet was a delay in obtaining or providing information about an abnormal NBS result and admission of the newborn to hospital. In the study from 95 centres across 21 European countries, over 60% of centres had commenced a low phenylalanine diet by 10 days of age. The highest results were observed in Western Europe (Germany and Austria), with 93% of centres reporting this, while the lowest results were observed in Eastern Europe (Latvia, Poland, Slovakia, Hungary, and Estonia), with 30% of centres reporting this [24]. In a comparable manner to PKU, approximately 50% of newborns from Group 2 (NBS non-PKU) were also placed on an appropriate dietary treatment before the 10th day of life, which is the time point considered in our observation as the optimal time to commence dietary treatment before symptoms of the disease appear.

A considerable number of studies [32,33,34] have demonstrated an improved prognosis in patients with MSUD diagnosed early in NBS (i.e., on the 5th–6th day of life), in whom, thanks to the rapid introduction of the BCAA-free diet, the time of exposure of the central nervous system to the toxic effects of metabolites of these amino acids was significantly reduced. The therapeutic benefits of immediately introducing a lysine- and tryptophan-limited diet in the first days of life have also been demonstrated in newborns with glutaric aciduria Type 1, diagnosed in NBS [35,36]. However, further studies are required to confirm the validity and outcomes of treatment initiated before the 10th day of life in non-PKU patients diagnosed by NBS.

During the diagnostic period, lactation disorders were observed more frequently in non-PKU groups than in those with classical phenylketonuria. Following the diagnosis of PKU, 26 (39%) newborns received human milk from a bottle to facilitate better control of phenylalanine intake. In situations where it was difficult to achieve metabolic control, dietitians recommended that mothers breastfeed, but in some cases, mothers made independent decisions. Interviews with mothers revealed that some of them hesitated to breastfeed for fear of harming their babies. Despite the medical staff’s explanations, they believed that they would feel safer feeding their babies expressed milk because it would allow for better control of food intake. A survey by Banta-Wright et al. [23] revealed that 20% of mothers who initiated breastfeeding after childbirth ceased this practice following a diagnosis of PKU in their child, while 22% breastfed for less than 1 month. Such psychological factors were also reported in a population of healthy infants. It can be concluded that the psychological factor determining the cessation of breastfeeding also concerned healthy infants in the Polish population. The most frequently indicated reason for the cessation of breastfeeding was a lack of milk (41%) [37], with the belief that there is not enough milk for the infant also being a significant factor [38].

A noteworthy observation in our study was the high percentage of infants with PKU (Group 1, 74%) and non-PKU (Group 2, 80%), diagnosed through NBS, who received human milk in the first month of life. In Group 1 (NBS-PKU), the proportion of infants who were breastfed (with or without intake control) versus those who were fed a measured amount of human milk from a bottle was almost 50:50. In contrast, in Group 2 (NBS non-PKU), the proportion was 60:40. A larger number of infants with non-PKU diseases diagnosed in NBS, i.e., before the onset of disease symptoms, allowed for more frequent feeding with human milk (from the breast or from a bottle) than in the case of SS. Among the infants in Group 2, the following proportions were observed: 11 of 12 from GA1, nine of twelve from IVA, two of three from HCU, six of seven from MSUD, one from PA, and four of five from MMA. For a period of no longer than the first four weeks of life, three out of five infants (Group 2) with urea cycle disorders (one with OTC, one with ASA, and one with Cytr.t.1) were fed with human milk from a bottle. By 3 months of age, the proportion of infants in Group 1 receiving human milk from a bottle had decreased by approximately 40%, while in Group 2, the decline was approximately 30%. However, it is notable that the decline was concentrated in infants fed from a bottle rather than directly from the breast. At this time, breast milk was still received in Group 2 (non-PKU NBS): six infants with GA1, eight with IVA, two with HCU, four with MSUD, one with PA, and three with MMA. Our observations indicate that the model of feeding from a bottle with expressed milk may have had an adverse effect on maintaining lactation and may have contributed to a faster transition to formula milk.

In Group 3, SS non-PKU, only six out of 16 infants received breast milk in the first month of life. Among these infants, four were diagnosed with a metabolic disorder: one had MSUD, two had Tyr t.1, and one had GA1. Two infants were not yet diagnosed, one with GA1 and one with Tyr t.1. The GA1 infant was diagnosed with a genetic abnormality within 3 months, while the Tyr t.1 infant was diagnosed by 5 months. In the remaining 10 newborns, breastfeeding was not possible due to the presence of acute clinical symptoms in the first days of life. These symptoms included feeding difficulties and a lack of milk production, which was the result of stress experienced by the mother.

A protein restriction in the diet was applied to infants in Group 2, with the exclusion of an amino acid mixture but with the inclusion of a protein-free formula. This was used for a period of 3 months, and it concerned four infants with IVA and three infants with MMA. In addition, infants diagnosed with IVA (two infants) and non-classic MSUD (two infants) in Group 2 remained on the HM + IF diet (with the biochemical control of the plasma amino acid level) without the modification of protein intake for the first 3 months of life.

An understanding of the composition of amino acids in human milk allows for the regulation of their supply and consumption in relation to their concentration in the blood. It is of paramount importance to maintain metabolic balance during dietary treatment. Based on the biochemical results, the order of consumption of breast milk or special formula (SF) was modified in our patients, in accordance with the procedure described by Huner et al. [27]. In the sixth month of life, there was a statistically significant decrease in the percentage of infants (across all groups) fed with human milk, from 72% to 27%.

In our group of infants with PKU, only eight of the 49 who received human milk continued to consume it from the breast without restriction for the first 6 months of life. Typically, after consuming a Phe-free formula, the infant was offered the breast without monitoring milk intake. However, the total number of feedings (HM plus SF) per day was often discussed with a dietitian. In general, infants received 8 to 12 feedings per day. Van Rijn et al. [39] described the similar approach to breastfeeding but with a set number of breast-feedings per day in a fixed schedule alternated with Phe-free formula. The amount of breast milk was not controlled by weight checks.

BF on demand without the use of special formulas is a viable option in the milder cases of IEM amino acids [26,27]. In our study, only infants in Group 2 (NBS non-PKU) were fed this way for 3–6 months, with eight infants with IVA, three infants with MSUD (no classical form) and two with MMA also included. One infant with ASA was fed this way for only a month. In the remaining patients with other aminoacidopathies, the practice of feeding on demand and without control of intake was not observed. Other authors similarly rarely recommended exclusive breastfeeding (without supplementation), and only in isolated cases of PA and ASA was it used successfully [40]. Recent studies by Buckingham et al. [41] demonstrated variable dietary practices used by registered dietitians regarding breastfeeding in infants with IEM. A lack of formal training programmes and/or dietetic standards in managing infants with inborn metabolic diseases may explain some of this variation.

It should be noted that this study is subject to certain limitations. As it is a retrospective study, it is not yet possible to determine the long-term outcomes of the effects with certainty. 

## 5. Conclusions

The diagnosis of inborn errors of metabolism in an infant is a significant source of stress for a mother who is required to commence “modified” breastfeeding of her child. Any form of human milk diet, with control of intake by weighing the child before and after feeding or control of the intake of expressed milk, involves the burden of additional activities related to it and may increase the mother’s anxiety and the risk of milk loss.

It is possible to maintain lactation and extend the period of feeding the infant with human milk in the first 6 months of life by breastfeeding on demand, under regular plasma biochemical monitoring every week in infants with PKU and at least every 2–4 weeks in non-PKU infants (especially in mild cases of IEM). This physiological method of infant feeding is beneficial for the mother and child with IEM during this challenging period of their lives.

It is recommended that breastfeeding be encouraged more frequently, provided that the type of IEM and its clinical presentation allow it, and close biochemical monitoring is possible. There is mounting evidence to suggest that breastfeeding is a safe practice for infants with PKU and non-PKU. However, there is a paucity of knowledge regarding the factors that influence breastfeeding. It is evident that psychological factors and the mother’s attitude are crucial elements in this context. Consequently, further research and special procedures are necessary to support the efforts of these mothers to continue breastfeeding after their infants are diagnosed with an inborn error of metabolism.

## 6. Home Messages

Early diagnosis and dietary treatment started without delay may increase the frequency of breastfeeding in infants with amino acid metabolism disorders.Breastfeeding after the intake of special formula for each inborn error of metabolism may be the best type of diet to help maintain breastfeeding for a longer period.Increased support from a qualified lactation specialist may reduce the percentage of women who cease breastfeeding.

## Figures and Tables

**Figure 1 nutrients-16-02544-f001:**
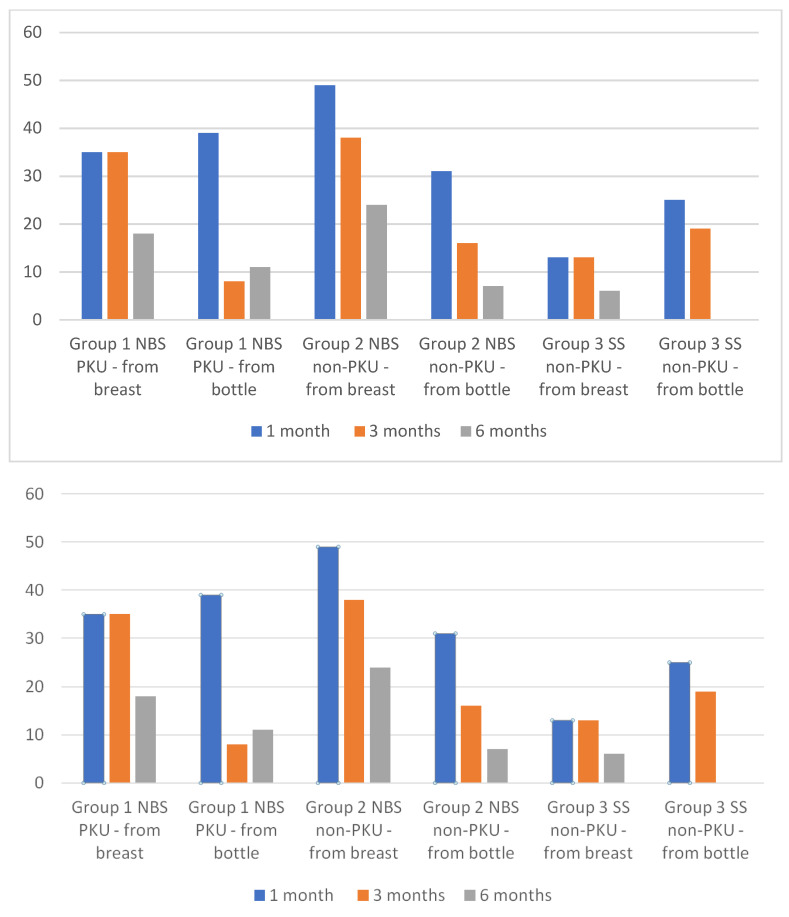
The proportion of infants who were breastfed and fed with expressed maternal milk at 1, 3, and 6 months of life.

**Table 1 nutrients-16-02544-t001:** Age at initiation of dietary treatment.

Initiation of Dietary Treatment	Group 1 NBS-PKU n = 66 (%)	Group 2 NBS non-PKU n = 45 (%)	Group 3 SS non-PKU n = 16 (%)
<10 days of life ≥10 days of life	31 (47%) 35 (53%)	21 (47%) 24 (53%)	0 16 (100%)

NBS PKU—patients diagnosed through newborn screening (NBS) with classic PKU; NBS non-PKU—patients diagnosed through newborn screening with: GA1-12, MSUD-7, PA-1, MMA-5, IVA-12, HCU-3, OTC-2, CIT t.1-2, and ASA-1; SS non-PKU—patients diagnosed through selective screening (SS) with: GA1-5, MSUD-3, PA-1, MMA-2, and TYR t.1-5

**Table 2 nutrients-16-02544-t002:** The proportion of infants who were fed or not fed with human milk (HM) from the breast or bottle at 1, 3, and 6 months of life.

Type of Feeding	1st Month				3rd Month				6th Month			
	Gr 1 NBS–PKU n = 66	Gr 2 NBS non-PKU n = 45	Gr 3 SS non-PKU n = 16	Total n = 127	Gr 1 NBS–PKU n = 66	Gr 2 NBS non-PKU n = 45	Gr 3 SS non-PKU n = 16	Total n = 127	Gr 1 NBS–PKU n = 66	Gr 2 NBS non-PKU n = 45	Gr 3 SS non-PKU n = 16	Total n = 127
Feeding with HM % of total group	49	36	6	91	28	24	5	57	19	14	1	34
	74	80	38	72	42	53	31	45	29	31	6	27
HM from breast % of total group	23	22	2	47	23	17	2	42	12	11	1	24
	35	49	13	37	35	38	13	33	18	24	6	19
HM from bottle % of total group	26	14	4	44	5	7	3	15	7	3	0	10
	39	31	25	35	8	16	19	12	11	7	0	8
None of HM % of total group	17	9	10	36	38	21	11	70	47	31	15	93
	26	20	62	28	58	47	69	55	71	69	94	73

NBS PKU—patients diagnosed through newborn screening (NBS) with classic PKU; NBS non-PKU—patients diagnosed through newborn screening with: GA1-12, MSUD-7, PA-1, MMA-5, IVA-12, HCU-3, OTC-2, CIT t.1-2, and ASA-1; SS non-PKU—patients diagnosed through selective screening (SS) with: GA1-5, MSUD-3, PA-1, MMA-2, and TYR t.1-5

**Table 3 nutrients-16-02544-t003:** Types of diet in the examined groups fed with human milk.

Months of Life	Type of Diet						
	HM + SF		HM + IF + SF		HM + Pf-F	HM	HM + IF
	with the Control of HM Intake	without the Control of HM Intake	with the Control of HM Intake	without the Control of HM Intake			
Group 1 NBS–PKU							
1 month, n = 49	26	8	15	0	0	0	0
3 months, n = 28	16	8	4	0	0	0	0
6 months, n = 19	8	8	3	0	0	0	0
Group 2 NBS non-PKU							
1 month, n = 36	7 GA1, 1 IVA, 2 HCU, 2 MSUD, 1 PA, 1 OTC, 1 CIT t.1	0	4 GA1, 1 MSUD, 1 MMA	0	4 IVA, 3 MMA	2 IVA *, 1 MSUD *	2 IVA *, 2 MSUD, 1 ASA
3 months, n = 24	5 GA1, 1 IVA, 1 HCU, 1 MSUD, 1 PA	0	1 GA1, 1 HCU, 1 MMA	0	4 IVA, 2 MMA	2 IVA, 1 MSUD *	1 IVA, 2 MSUD *
6 months, n = 14	2 GA1, 1 PA	0	1 GA1, 2 HCU	0	2 IVA, 1 MMA	2 IVA, 1 MSUD *	1 IVA, 1 IVA *
Group 3 SS non-PKU							
1 month, n = 6	2	0	2	0	0	2 *	0
3 months, n = 5	1	0	3	0	0	1 *	0
6 months, n = 1	0	0	0	0	0	1 *	0

* without the control of the human milk intake; ASA—Argininosuccinic Aciduria; CIT t.1—Citrullinemia Type 1; GA 1—Glutaric Aciduria Type 1; HCU—Homocystinuria; IVA—Isovaleric Aciduria; MMA—Methylmalonic Aciduria; MSUD—Maple Syrup Urine Disease; OTC—Ornithine Transcarbamylase Deficiency; PA—Propionic Aciduria; PKU—Phenylketonuria. NBS PKU—patients diagnosed through newborn screening (NBS) with classic PKU; NBS non-PKU—patients diagnosed through newborn screening with: GA1-12, MSUD-7, PA-1, MMA-5, IVA-12, HCU-3, OTC-2, CIT t.1-2, and ASA-1; SS non-PKU—patients diagnosed through selective screening (SS) with: GA1-5, MSUD-3, PA-1, MMA-2, and TYR t.1-5

## Data Availability

Data are available upon request.

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
