# Peer review of "Breastfeeding in PKU and Other Amino Acid Metabolism Disorders—A Single Centre Experience"

_nutrients, 2024, doi:10.3390/nu16152544_

Round 1

Reviewer 1 Report

Comments and Suggestions for Authors

I have read this paper with interest, and confirm that the data on PKU related breastfeeding confirms a significant number of other observational studies. However, to the best of my understanding, there are no RCT studies on this topic. In fact, the novelty rather related to these non-PKU amino-acid related metabolic syndrome.

I highly recommend the authors to reconsider the title, as the non PKU cases do have other aminoacid related metabolic syndromes, and therefore are not just non PKU cases.

On the analysis, causality is difficult to assess because the retrospective design. I therefore would like to suggest to rather use association type of language. This is already somewhat reflected in the discussion section, where the authors reflect on the associations/driver of breastfeeding versus bottle-provided human milk.

Introduction: there are also advantages to the mother to breastfeed versus formula feeding. I would recommend to add this.

Methods: we miss a reflection on the ‘targets’ (amino acid concentrations) during the follow up. We do need more details on this aspect of the paper. The same holds true when you refer to the ‘selected screening’ practices. The readership simply needs more detailed description on this.

108-112: in my ‘external’ assessment, all cases remain under the control of the institute ? but you likely want this statement in the paper because it is perceived meaningful. What was your motivation, and please explain this, or remove.

144: human milk volume, but how to assess this in breastfed infants, was this based on a fixed value, or sequential weighting ?

215: human milk is also associated with improved maturation of the CNS (myelinization), so that I suggest to add this.

Finally, what are the potential take home messages or action points of your analysis

Reviewer 2 Report

Comments and Suggestions for Authors

Review for the manuscript “ Breastfeeding in PKU and non-PKU infants – a single centre experience ”

 This study tried to analyze breastfeeding and the degree of contribution of breast milk to the feeding of infants with inborn errors of amino acid metabolism  (IEM) before and after the introduction of expanded newborn screening. The authors investigated 127 infants with phenylketonuria (PKU), 45 with other IEM (non-PKU), both groups diagnosed in newborn screening (NBS), and 16 non-PKU diagnosed in selective screening (SS). They found that maintaining lactation and extending the period of feeding the infant with human milk in the first six months of life is possible by breastfeeding on demand, preferably weekly in PKU infants, and at least every 2-4 weeks in infants with other IEM.

How did the authors establish the dietary treatment for group 1 and 2 prior to the 10th day of life and for group 3 after the 10th day of their life ?

The authors mentioned that they determine biochemical parameters, such as amino acid levels, phenylalanine levels. Please described the methods used. Also provide information regarding the provenience of the reagents. How did they determine argininosuccinic Aciduria, Citrullinemia Type 1, Glutaric Aciduria Type 1, Homocystinuria, Isovaleric Aciduria, Methylmalonic Aciduria, Maple Syrup Urine Disease, Ornithine Transcarbamoylase Deficiency, Propionic Aciduria, Phenylketonuria. 

Round 2

Reviewer 1 Report

Comments and Suggestions for Authors

the comments have been well addressed, suggest to accept